# Improving Uncertainty Quantification in Large Language Models via Semantic Embeddings

## Abstract

Accurately quantifying uncertainty in large language models (LLMs) is crucial for their reliable deployment, especially in high-stakes applications. Current state-of-the-art methods for measuring semantic uncertainty in LLMs rely on strict bidirectional entailment criteria between multiple generated responses and also depend on sequence likelihoods. While effective, these approaches often overestimate uncertainty due to their sensitivity to minor wording differences, additional correct information, and non-important words in the sequence. We propose a novel approach that leverages semantic embeddings to achieve smoother and more robust estimation of semantic uncertainty in LLMs. By capturing semantic similarities without depending on sequence likelihoods, our method inherently reduces any biases introduced by irrelevant words in the answers. Furthermore, we introduce an amortised version of our approach by explicitly modelling semantics as latent variables in a joint probabilistic model. This allows for uncertainty estimation in the embedding space with a single forward pass, significantly reducing computational overhead compared to existing multi-pass methods. Experiments across multiple question-answering datasets and frontier LLMs demonstrate that our embedding-based methods provide more accurate and nuanced uncertainty quantification than traditional approaches.

## 1 Introduction

Large Language Models (LLMs) have revolutionised natural language processing (see e.g. The Gemini Team,, 2023; Touvron et al., 2023; OpenAI, 2023; Brown et al., 2020), achieving state-of-the-art performance across a wide variety of tasks including question-answering. As these models are increasingly deployed in critical domains like healthcare (Singhal et al., 2023) and law (Weiser, 2023) ensuring their reliability and trustworthiness has become imperative. A significant challenge in this context is the phenomenon of "hallucinations"—instances where LLMs generate fluent and coherent responses that are factually incorrect or misleading (Ji et al., 2023; Filippova, 2020; Maynez et al., 2020; Tian et al., 2024).

Uncertainty quantification (UQ) methods like Bayesian inference (Wilson & Izmailov, 2020), ensemble methods (Lakshminarayanan et al., 2017), and Monte Carlo dropout (Gal & Ghahramani, 2016) have been extensively studied in traditional neural networks to enhance model reliability by providing confidence measures in predictions. However, applying these traditional UQ methods to LLMs faces challenges due to the open-ended nature of free-form natural language generation (Kuhn et al., 2023). The core issue lies in the fundamental mismatch between traditional UQ approaches, which typically estimate uncertainty in output probabilities, and the semantics (meaning) space of language generation in LLMs.

For example, consider the following scenario of two responses generated for the same query: "London is the biggest city in the UK". "The largest city in the UK is London". In this scenario, the output token probabilities will include uncertainty about the syntax and choice of words (e.g., using "biggest" or "largest"), along with the uncertainty about the underlying semantic content of the response. Traditional UQ techniques focusing on token probabilities conflate these different sources

of uncertainty, making it challenging to isolate semantic uncertainty, which is critical for assessing the reliability of the generated information.

Semantic entropy (Kuhn et al., 2023) aims to isolate semantic uncertainty by sampling multiple answers from the LLM for a given prompt. It does so by clustering the generations into sets of equivalent semantics and then estimating uncertainty in the space of identified semantics. The premise is that higher semantic uncertainty leads to more diverse meanings in the generated responses, while lower uncertainty results in more semantically consistent responses. In order to cluster semantically equivalent answers, semantic entropy leverages a strict bidirectional entailment criterion which, as we show in this work, can be sensitive to minor variations in wording, additional correct information, or non-essential words in the generated responses. Such sensitivity can lead to an overestimation of semantic uncertainty. Additionally, semantic entropy requires multiple forward passes, which can limit its practicality in production environments where low latency is essential and for larger LLMs where each forward pass incurs substantial computational expenses.

To address these challenges, we make the following key contributions:

- **Semantic Embedding Uncertainty (SEU)**: We introduce SEU, by leveraging the average pairwise cosine similarity of full response embeddings SEU avoids the issues associated with using bi-directional entailment as a criterion for clustering semantically equivalent responses (see sections 3 and 4).

- **Amortised SEU**: We present an amortised version that models semantics as latent variables in a joint probabilistic model. This allows for the estimation of posterior uncertainty in the latent semantics within a single forward pass, alongside the response generation, significantly reducing computational overhead and enhancing practicality for deployment in production environments (see sections 5 and 6).

## 2 BACKGROUND

We first provide a concise summary of Semantic Entropy and detail its limitations which form the motivation of our proposed approaches.

### 2.1 SEMANTIC ENTROPY

**Semantic Entropy** (SE) is a measure of uncertainty in sequences generated by language models (Kuhn et al., 2023). The central idea is that if a language model is unsure about how to answer a specific question, it will produce responses that are different in wording *and* semantics across multiple generations when the model is given the same input. SE groups semantically similar responses and calculates the entropy based on the variety of distinct meanings found in the output responses. Specifically, the key steps involved are: (i) sample $M$ output sequences $\{\mathbf{s}_1, \ldots, \mathbf{s}_M\}$ from the language model's predictive distribution $p(\mathbf{s}|\mathbf{x})$ given an input $\mathbf{x}$; (ii) cluster the sampled sequences into $K$ semantic equivalence classes $C = \{c_1, \ldots, c_K\}$ using a bidirectional entailment algorithm. Two sequences $\mathbf{s}_i$ and $\mathbf{s}_j$ are considered semantically equivalent *if and only if* a natural language inference model classifies their mutual relationship as entailment in both directions; (iii) estimate the probability of each semantic cluster $p(c_k|\mathbf{x}) = \sum_{\mathbf{s} \in c_k} p(\mathbf{s}|\mathbf{x})$; and (iv) compute the entropy over semantic clusters: $\text{SE}(\mathbf{x}) = -\sum_{k=1}^{K} p(c_k|\mathbf{x}) \log p(c_k|\mathbf{x})$.

#### 2.1.1 BIDIRECTIONAL ENTAILMENT AND ITS LIMITATIONS

We focus on step (ii), where bidirectional entailment is used to identify distinct semantic clusters among the $M$ responses. This strict criterion, however, can be overly sensitive to minor variations in wording, additional correct information, or non-essential words. This issue is illustrated by several examples listed in Table 1.

**Example 1: Generality Mismatch in Responses** Both responses correctly state that mitochondria produce energy. However, bidirectional entailment fails because the first response uses "produce energy for the cells", while the second uses "provides energy to cells in the body". The addition of "in the body" in the second response making it a less general statement than the first leads to a *False* classification despite a high cosine similarity of 0.974.

Table 1: Comparison of bidirectional entailment and cosine similarity for assessing semantic equivalence. DeBERTaLarge (He et al., 2021) is used to predict entailment as used in Kuhn et al. (2023), and the inputs to the cosine similarity are obtained using sentence-BERT (Reimers & Gurevych, 2019).

| Context | Responses | Bidirectional Entailment | Cosine Similarity |
|---|---|---|---|
| What is the primary function of the mitochondria in cells? | 1. The mitochondria produce energy for the cells. 2. Mitochondria provides energy to cells in the body. | False | 0.974 |
| What happens when you heat ice? | 1. Heating ice will eventually boil after becoming water. 2. When ice is heated, it melts into water before boiling. | False | 0.893 |
| What do mammals have in common? | 1. Mammals are warm-blooded and have hair or fur. 2. All mammals (like humans and dogs) are warm-blooded creatures with hair. | False | 0.927 |

**Example 2: Phrasing Variations**    Both responses accurately describe the process of heating ice, but bidirectional entailment fails due to different information orders ("eventually boil" vs. "before boiling"). These temporal differences in phrasing result in a *False* classification, despite a cosine similarity of 0.893.

**Example 3: Additional Correct Information**    Both responses identify key traits of mammals, but bidirectional entailment fails because the second response includes additional correct information ("like humans and dogs") not present in the first. These additions and wording differences lead to a *False* classification, despite a high cosine similarity of 0.927.

As the above examples demonstrated, natural language is inherently varied, and strict binary classifications do not account for the nuances and gradations in meaning that often occur in human language. Bidirectional entailment treats semantic equivalence as a binary condition: responses either fully entail each other or they do not. This strictness can lead to it being over-sensitive to minor variations as shown in our examples, small differences in phrasing or the inclusion of additional but correct information can cause bidirectional entailment to fail, even when the core meaning is preserved. In contrast, the high cosine similarity scores across all examples suggest that these responses are indeed very close in semantic space. To address this limitation and more robustly quantify semantic uncertainty, we propose using the average pairwise cosine similarity of the generated responses. This approach can capture semantic closeness more flexibly, allowing for minor variations in wording or additional correct information without overly penalising the uncertainty estimate. We detail this proposed method in the following section.

While the average pairwise cosine similarity approach addresses the limitations of binary semantic classifications, it still has limited practical applicability. Both SE and the proposed method, require multiple forward passes through the language model. This requirement significantly limits their practicality in production environments, especially for large LLMs where each forward pass incurs a substantial computational cost. To this end, we propose a novel method in Section 5 that treats the semantics of the response as latent variables in a joint probabilistic model. Our approach employs amortised inference over the semantics of the full response, allowing us to estimate the LLM's uncertainty about the semantics of its entire response (i.e., the complete sequence of tokens) in a single forward pass. This method drastically reduces the computational overhead required to estimate semantic uncertainty while preserving the benefits of comparing semantic embeddings using cosine similarity.

## 3    SEMANTIC EMBEDDING UNCERTAINTY

To overcome the limitations of bidirectional entailment in measuring semantic uncertainty, we propose *semantic embedding uncertainty* (SEU), a novel approach based on the **average pairwise cosine similarity** of the generated responses' embeddings. This method leverages continuous semantic representations to capture nuanced meanings more precisely, offering a robust measure of semantic uncertainty in language model outputs.

Similar to Semantic Entropy, given an input $\mathbf{x}$, we generate $M$ output sequences $\{\mathbf{s}_1, \mathbf{s}_2, \ldots, \mathbf{s}_M\}$ from the language model's predictive distribution $p(\mathbf{s}|\mathbf{x})$. We then obtain vector embeddings $\{\mathbf{e}_1, \mathbf{e}_2, \ldots, \mathbf{e}_M\}$ for each sequence using a pretrained embedding model $\phi(\mathbf{s})$, such as a transformer-based sentence encoder (Reimers & Gurevych, 2019). The semantic uncertainty is quantified by computing the negative average pairwise cosine similarity between the embeddings:

$$\text{SEU}(x) = 1 - \frac{2}{M(M-1)} \sum_{i=1}^{M-1} \sum_{j=i+1}^{M} \cos(\mathbf{e}_i, \mathbf{e}_j), \tag{1}$$

where $\cos(\mathbf{e}_i, \mathbf{e}_j)$ is the cosine similarity between embeddings $\mathbf{e}_i$ and $\mathbf{e}_j$, $\cos(\mathbf{e}_i, \mathbf{e}_j) = \frac{\mathbf{e}_i \cdot \mathbf{e}_j}{\|\mathbf{e}_i\| \|\mathbf{e}_j\|}$, $\|\mathbf{e}\|$ denotes the Euclidean norm of vector $\mathbf{e}$, and $\mathbf{e}_i \cdot \mathbf{e}_j$ represents the dot product between vectors $\mathbf{e}_i$ and $\mathbf{e}_j$.

The proposed approach relies on high-quality embedding models to map semantically similar sentences to nearby points in the embedding space (e.g. Mikolov et al., 2013; Pennington et al., 2014; Reimers & Gurevych, 2019). Intuitively, cosine similarity quantifies the angle between vectors in the high-dimensional embedding space, serving as a measure of their semantic alignment. By aggregating pairwise similarities across all generated responses, we capture the overall semantic coherence of the model's outputs. If the language model is certain about the response to input $x$, the generated responses will be semantically similar, leading to high cosine similarity scores and a low semantic uncertainty $\text{SEU}(x)$. Conversely, if the model is uncertain, the responses will be more diverse semantically, resulting in lower cosine similarity scores and a higher $\text{SEU}(x)$.

The proposed approach offers two key advantages over bidirectional entailment. **First**, unlike the *binary* outcome of bidirectional entailment—which rigidly classifies responses as either semantically equivalent or not—cosine similarity provides a *continuous* metric. This allows for a nuanced assessment of semantic closeness between responses. While bidirectional entailment may fail to recognise near-equivalent meanings due to minor differences (thereby assigning a value of zero similarity), cosine similarity captures the *degree* of similarity between responses. This continuous spectrum more accurately reflects the gradations in human language understanding. **Second**, as shown above, bidirectional entailment is highly sensitive to syntactic variations, paraphrasing, and the inclusion of additional relevant information, often resulting in false negatives when determining semantic equivalence. In contrast, cosine similarity focuses on the underlying *semantic content* rather than exact entailment. This makes it less sensitive to linguistic variability, such as differences in syntax or phrasing.

## 4 Empirical Evaluation of SEU

In this section, we empirically evaluate our proposed SEU method against existing uncertainty estimation techniques. Our goal is to demonstrate that SEU provides a more accurate and robust measure of semantic uncertainty in language model outputs, particularly in the context of open-domain question answering.

### 4.1 Experimental Setup

**Models** To align with modern practices of using instruction fine-tuned LLMs for chat purposes, we employ Llama-3.1-8B-Instruct (Dubey et al., 2024), Mistral-7B-Instruct (Jiang et al., 2023), and Phi-3.5-mini-instruct (Abdin et al., 2024) as our base models. For each model-dataset combination, we generate 5 responses per question (that is $M = 5$) at a temperature of 0.5. This temperature was recommended as the optimal temperature for SE in previous work (Kuhn et al., 2023). Additionally, we prompt Llama and Phi models with "Answer the following question as briefly as possible", and Mistral with "Answer the following question briefly using a few words" to match the short-answer format of our datasets.

**Datasets** We evaluate our proposed Semantic Embedding Uncertainty (SEU) method on three challenging question-answering datasets: TriviaQA (Joshi et al., 2017), NQ Open (Kwiatkowski et al., 2019; Lee et al., 2019), and the natural question subset of the FLAN collection (Longpre et al., 2023, Flan QA) [1]. TriviaQA offers a large set of trivia questions both with and without relevant context, NQ

---

[1] https://huggingface.co/datasets/Muennighoff/flan

Open provides real user queries requiring short answers, and Flan QA also includes short question-answer pairs specifically designed for instruction-tuning language models. These datasets provide a diverse range of questions and answers, allowing us to assess the robustness of our method across various domains and question types. The selection of these datasets enables us to evaluate SEU's performance on both traditional QA tasks as well as more recent instruction-following scenarios.

**Baselines** Following the evaluation by Kuhn et al. (2023), we compare our SEU method against the following baselines: Predictive Entropy which is just the average predictive entropy of all the tokens in the sequence, Length-normalised Predictive Entropy which is the joint log-probability of each sequence divided by the length of the sequence (Malinin & Gales, 2021)[2], and Semantic Entropy (Kuhn et al., 2023). For predicting bidirectional entailment in the Semantic Entropy baseline, we use DeBERTaLarge (He et al., 2021) as used in Kuhn et al. (2023), while for our SEU method, we utilise the sentence-BERT (Reimers & Gurevych, 2019) for semantic embeddings. Following prior work (Kuhn et al., 2023; Duan et al., 2023), we use the Area Under the Receiver Operating Characteristic curve (AUROC) as our primary evaluation metric. This metric treats uncertainty estimation as the problem of predicting whether to rely on a model's generation for a given context. We evaluate the correctness of our model's generations using a fuzzy matching criterion based on the Rouge-L score, considering an answer correct if its Rouge-L score with respect to the reference answer is larger than 0.3. Our experimental procedure involves computing uncertainty scores using our proposed SEU method and the baselines for each generated response, evaluating their correctness, and calculating AUROC scores. All experiments were done using one NVIDIA A100 GPU.

## 4.2 RESULTS

Our empirical evaluation demonstrates the effectiveness of the proposed Semantic Embedding Uncertainty (SEU) method across different models and datasets. We present our findings in two parts: a comparative analysis of uncertainty estimation methods and an in-depth examination of the trade-off between false positive rate (FPR) and true positive rate (TPR).

### 4.2.1 COMPARATIVE ANALYSIS OF UNCERTAINTY ESTIMATION METHODS

Figure 1 presents the AUROC scores for different uncertainty estimation methods across three models (Llama-3.1-8B-Instruct, Phi-3.5-Instruct, and Mistral-7B-Instruct) and three datasets (TriviaQA, NQ Open, and Flan QA). We note the proposed SEU method consistently outperforms or matches the performance of other uncertainty estimation methods across all model-dataset combinations. Specifically, while the relative performance of methods varies slightly across models, SEU maintains its advantage, suggesting robustness to model architecture differences. The performance patterns differ across datasets, with all methods generally performing better on TriviaQA compared to NQ Open and Flan QA. Crucially, SEU consistently outperforms Semantic Entropy, supporting our hypothesis that the latter may overestimate uncertainty due to its sensitivity to minor linguistic variations.

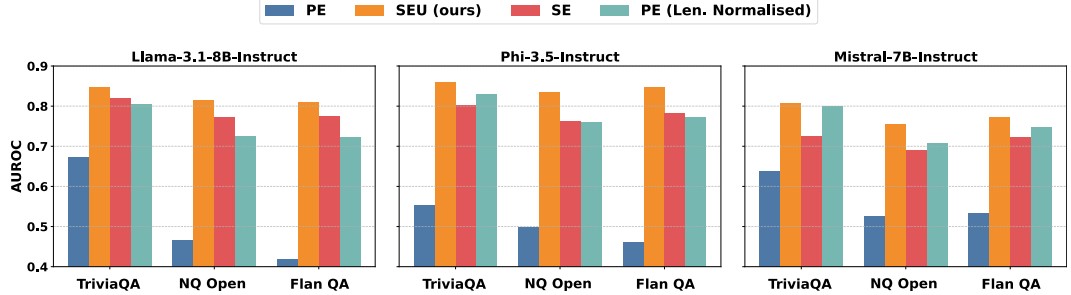

Figure 1: Comparison of SEU method against baselines across different models and datasets.

---

[2]This technically should be called length-normalised log-likelihood, but we follow prior work on using this name here.

### 4.2.2 ANALYSIS OF FALSE POSITIVE RATE AND TRUE POSITIVE RATE TRADE-OFF

To further investigate the performance difference between SEU and Semantic Entropy, we analyse the False Positive Rate (FPR) and True Positive Rate (TPR) at the optimal Youden's J statistic point for the NQ Open dataset. A "positive" case refers to an instance where the model's response is correct. Table 2 presents these results. Notably, SEU consistently achieves a higher TPR compared to Semantic Entropy across all models. This indicates that SEU is more effective at identifying the cases when the LLM is confident about the underlying semantics. The improved TPR of SEU comes with a slight increase in FPR. However, the gain in TPR (ranging from 0.0897 to 0.1785) outweighs the increase in FPR (ranging from 0.0460 to 0.1037), resulting in better overall performance as reflected in the AUROC scores.

Table 2: Comparison of Semantic Embedding Uncertainty and Semantic Entropy on NQ Open Dataset at Optimal Youden's J Statistic Point

| Model | SEU (Ours) | | SE | |
|---|---|---|---|---|
| | FPR [↓] | TPR [↑] | FPR [↓] | TPR [↑] |
| Llama 3.1 8B | 0.2608 | 0.7767 | 0.1655 | 0.6543 |
| Phi 3.5 | 0.2198 | 0.7571 | 0.1738 | 0.6674 |
| Mistral 7B | 0.3293 | 0.7353 | 0.2256 | 0.5568 |

This empirical evidence supports our argument that Semantic Entropy may overestimate uncertainty as demonstrated by its significantly lower TPR. Lower TPR suggests that Semantic Entropy is classifying more cases as uncertain, even when the model's response is correct. The higher TPR of SEU suggests that it captures a more nuanced view of semantic similarity, allowing it to identify truly uncertain cases more accurately.

## 5 AMORTISED SEMANTIC EMBEDDING UNCERTAINTY

While our proposed Semantic Embedding Uncertainty (SEU) method demonstrates superior performance in uncertainty estimation across various models and datasets, it shares a significant limitation with Semantic Entropy: computational inefficiency. Both SEU and Semantic Entropy require multiple forward passes through the language model to generate a set of responses for each input, which can be prohibitively expensive, especially for large language models in production environments. To this end, we present amortised SEU (ASEU) to tackle the challenge of estimating semantic uncertainty in a single forward pass. The goal is to represent the semantics of a sequence as latent variables and spend a small effort to finetune and obtain an amortised approximate posterior over them to bypass the need for an external paragraph or sentence embedding model at test time.

### 5.1 LATENT SEMANTIC MODEL

Suppose we have a training set of $N$ sequences and $\mathbf{x}_n = (x_{n,1}, x_{n,2}, \ldots, x_{n,T})$ is the $n$-th sequence that has $T$ tokens. We assume there is a latent vector $\mathbf{z}_n \in \mathbb{R}^D$ that captures the semantic of the $n$-th sequence and that the embeddings $\mathbf{e}_n \in \mathbb{R}^D$ of this sequence can be computed using an external, pretrained embedding model. The joint distribution over the latent semantic and observed embeddings is defined as follows,

$$p(\{\mathbf{e}_n, \mathbf{z}_n\}_{n=1}^N | \omega) = \prod_{n=1}^N p(\mathbf{z}_n) p(\mathbf{e}_n | \mathbf{z}_n, \omega),$$

We choose a standard normal prior over $\mathbf{z}_n$, $p(\mathbf{z}_n) = \mathcal{N}(\mathbf{z}_n; \mathbf{0}, \mathbf{I}_D)$ and $p(\mathbf{e}_n | \mathbf{z}_n, \omega) = \mathcal{N}(\mathbf{e}_n; \mathrm{NN}_\omega(\mathbf{z}_n), \sigma_e^2 \mathbf{I}_D)$, where $\mathrm{NN}_\omega$ is a small neural network with parameters $\omega$. It is worth noting that while modelling $\mathbf{z}_n$ at every time step is possible, this would involve specifying a dynamic prior mapping from $\mathbf{z}_{n,t-1}$ to $\mathbf{z}_{n,t}$ and computing the embeddings for all subsequences $(x_{n,1}, x_{n,2}, \ldots, x_{n,t})$. We opt for simplicity and pick a global $\mathbf{z}$ for the whole sequence.

## 5.2 APPROXIMATE INFERENCE

Readers familiar with latent variable modelling might have noted similarity between the model above and Gaussian latent variable models in Kingma & Welling (2014); Rezende et al. (2014). The most natural next step for inference would be to impose an approximate posterior over $\mathbf{z}_n$, $q(\mathbf{z}_n|\mathbf{e}_n, \psi)$, that mirrors that of the exact posterior, $p(\mathbf{z}_n|\mathbf{e}_n, \omega)$. While this is arguably the most accurate approach, computing $\mathbf{z}$ for a new sequence at test time requires access to the embedding $\mathbf{e}$, which we seek to avoid. To sidestep this, we posit the variational Gaussian distribution over $\mathbf{z}$, $q(\mathbf{z}_n|\mathbf{x}_n, \psi, \theta) = \mathcal{N}(\mathbf{z}_n; \mu_n, \Sigma_n)$, where $\mu_n$ and $\Sigma_n$ are outputs of a fully-connected neural network, parameterised by $\psi$. This network takes as input the representation of $\mathbf{x}_n$ provided by the language model, that is parameterised by $\theta$. This parameterisation allows us to obtain a distribution over $\mathbf{z}$ using the same backbone as used for modelling $\mathbf{x}$. Equipped with the model and variational distribution specifications, we now wish to minimise the KL divergence between the approximate posterior and the true posterior, $\mathrm{KL}[q(\mathbf{z}_n|\mathbf{x}_n, \psi, \theta) \ || \ p(\mathbf{z}_n|\mathbf{e}_n, \omega)]$, or equivalently, minimising the negative lower bound to the log marginal likelihood $\log p(\{\mathbf{e}_n\}_{n=1}^N|\omega)$, $\mathcal{L}(\theta, \omega, \psi) = \sum_n \mathcal{L}_n(\theta, \omega, \psi)$, where

$$
\begin{aligned}
\mathcal{L}_n(\theta, \omega, \psi) &= \int_{\mathbf{z}_n} q(\mathbf{z}_n|\mathbf{e}_n, \psi, \theta) \left[ \log q(\mathbf{z}_n|\mathbf{e}_n, \psi, \theta) - \log p(\mathbf{e}_n, \mathbf{z}_n|\omega) \right] \\
&= \mathrm{KL}[q(\mathbf{z}_n|\mathbf{e}_n, \psi, \theta) \ || \ p(\mathbf{z}_n)] - \int_{\mathbf{z}_n} q(\mathbf{z}_n|\mathbf{x}_n, \psi, \theta) \log p(\mathbf{e}_n|\mathbf{z}_n, \omega).
\end{aligned}
$$

The above objective is intuitive: we want to fine-tune the language model and optimise the model and variational parameters such that the language model's representation helps reconstruct the embedding of the full sequence. Additionally, $\theta$ can be kept fixed if a pre-trained language model is already available or simultaneously fine-tuned using the negative log-likelihood of $\mathbf{x}$ as in conventional autoregressive language modelling.

## 5.3 UNCERTAINTY ESTIMATION AT TEST TIME

As the variational approximation is trained to approximately imitate the sequence embedding, it can be leveraged to estimate the semantic uncertainty similarly to the SEU method proposed earlier. At each step $t$ during response generation, we draw $K$ samples $\{\mathbf{z}_{t,1}, \ldots, \mathbf{z}_{t,K}\}$ from the approximate posterior $q(\mathbf{z}_t|\mathbf{x}_t, \psi, \theta)$, where $\mathbf{x}_t$ are the prompt and the tokens generated so far. We then compute the average pairwise cosine similarity between these samples:

$$
S_t = \frac{2}{K(K-1)} \sum_{j=1}^{K-1} \sum_{k=j+1}^{K} \cos(\mathbf{z}_{t,j}, \mathbf{z}_{t,k})
$$

where $\cos(\mathbf{z}_{t,j}, \mathbf{z}_{t,k})$ denotes the cosine similarity between samples $\mathbf{z}_{t,j}$ and $\mathbf{z}_{t,k}$. After generating the complete response of length $T$, we calculate the raw ASEU score: $\mathrm{ASEU}_{\mathrm{raw}} = 1 - \mathrm{median}\{S_1, \ldots, S_T\}$. The intuition behind this approach is that if the LLM is uncertain about the latent semantics of future tokens, the sampled embeddings at each step will have lower average similarity compared to cases where the LLM is more certain. Taking the median across all steps yields a robust measure of the overall semantic uncertainty for the entire response. To account for the impact of response length on uncertainty, we apply length normalization, defining our final ASEU score, with higher values indicating greater uncertainty and lower values indicating greater certainty. This normalization step is crucial as it mitigates potential bias towards longer responses, which might accumulate more uncertainty simply due to their length. We also found the proposed approach is more robust than using the entropy of the variational approximation.

## 6 EMPIRICAL EVALUATION OF AMORTIZED SEU

In this section, we empirically evaluate our proposed amortized Semantic Embedding Uncertainty (ASEU) method. Unlike the previous multi-pass setting, we now focus on estimating uncertainty in a single forward pass, which is crucial for practical applications in production environments.

### 6.1 EXPERIMENTAL SETUP

**Models:** We use the same three models as in the previous experiments: Llama-3.1-8B-Instruct, Phi-3.5-Instruct, and Mistral-7B-Instruct. However, for this evaluation, we fine-tune these models to optimize the variational objective presented in section 5.2.

**Fine-tuning:** To learn the approximate posterior distribution $q(\mathbf{z}|\mathbf{x}, \psi, \theta)$ and parameters $\theta$ and $\omega$, we fine-tune the LLMs on the TriviaQA dataset. We chose TriviaQA for fine-tuning due to its size and diverse coverage of question-answering tasks and its focus on short answers, which aligns with the paper's emphasis so far. This fine-tuning process allows the models to leverage the underlying language model to estimate semantic uncertainty in a single forward pass.

**Baselines:** In the single forward pass setting, we compare our ASEU method against the length-normalized predictive entropy of a single forward pass response. This baseline is chosen as it is the most relevant uncertainty estimation method that can be computed in a single pass.

### 6.2 RESULTS AND DISCUSSION

Figure 2 presents the AUROC scores for our ASEU method and the length-normalized predictive entropy baseline across the three models (Llama-3.1-8B-Instruct, Phi-3.5-Instruct, and Mistral-7B-Instruct) and two datasets (NQ Open and Flan QA). We also compare the armotised uncertainty estimates with SEU and other methods that require multiple generations in figure 3. We note that ASEU consistently outperforms the length-normalized predictive entropy baseline across all model-dataset combinations, suggesting it captures more meaningful uncertainty information. While ASEU generally doesn't match the performance of multi-pass SEU method, it achieves comparable results for the Mistral model. The performance gap between ASEU and multi-pass SEU varies across models and datasets, but the computational efficiency gained through single-pass estimation makes ASEU more suitable for real-world applications, especially in production environments where multiple forward passes are infeasible.

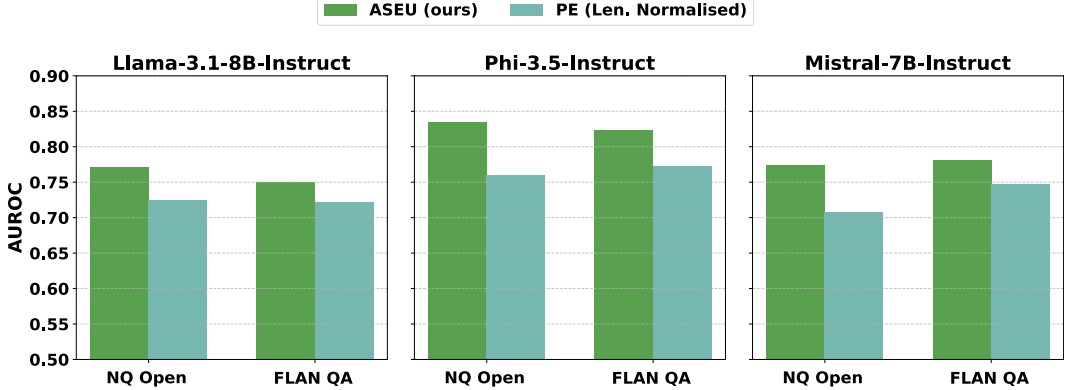

Figure 2: Comparison of amortised SEU method against log-likelihood in a single forward pass setting across different models and datasets.

### 6.3 ANALYSIS OF LEARNT LATENT EMBEDDINGS

To demonstrate that the latent embeddings of our amortized model carry meaningful semantic information, we examined the cosine similarities between the means of the variational distributions given semantically related queries and presented the results in Table 3. The perfect similarity between queries about England's capital and the UK's biggest city reflects their close relationship (both referring to London). The lower but consistent similarity (0.83) between these queries and a question about Australia's capital shows that the embeddings capture both the semantic structure of the questions (asking about capital/major cities) and the distinction between different locations. This suggests that the learnt embeddings after finetuning the LLMs encodes semantic relationships.

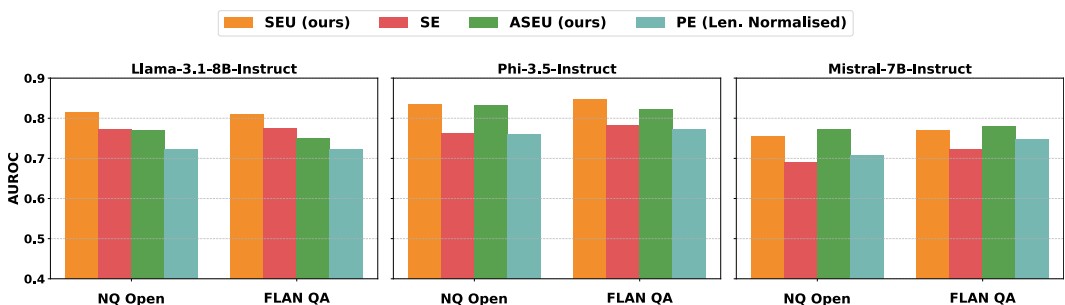

Figure 3: Comparison of amortised SEU method against techniques requiring multiple forward passes across different models and datasets.

Table 3: Cosine Similarities Between Predicted Embeddings of the following Query Embeddings

| Query 1 | Query 2 | Cosine Similarity |
|---|---|---|
| What is the capital of England? | What is the biggest city in the UK? | 1.00 |
| What is the capital of England? | What is the capital of AUS? | 0.83 |
| What is the biggest city in the UK? | What is the capital of AUS? | 0.83 |

## 7 RELATED WORKS

**Hallucinations in LLMs:** The challenge of hallucination detection in LLMs has become increasingly important as these models are deployed in real-world applications. Various benchmarks have been developed to evaluate this phenomenon, including TruthfulQA (Lin et al., 2021), FactualityPrompt (Lee et al., 2022), FActScore (Min et al., 2023), HaluEval (Li et al., 2023a), and FACTOR (Muhlgay et al., 2023). Early research on hallucinations primarily focused on issues in summarization tasks, where models would generate content unfaithful to the source text (Maynez et al., 2020; Durmus et al., 2020; Wang et al., 2020). This work laid the foundation for understanding the broader challenge of hallucinations in LLMs.

**Uncertainty Estimation Approaches:** A significant body of work has explored methods to estimate uncertainty in LLM outputs. Many of these approaches rely on comparing multiple model generations or outputs by leveraging additional LLMs or by using the same LLM (Duan et al., 2023; Chen & Mueller, 2023; Manakul et al., 2023; Mündler et al., 2023). The field has seen a variety of innovative techniques, including those proposed by Kadavath et al. (2022), Mitchell et al. (2022), and Xu et al. (2022), which leverage different aspects of model behaviour to gauge uncertainty.

**Knowledge Integration Methods:** Another line of research focuses on integrating external knowledge to verify and improve the factual accuracy of LLM outputs. The RARR framework (Gao et al., 2023) uses search engines for knowledge retrieval and correction. Similarly, the Verify-and-Edit approach (Zhao et al., 2023) leverages external information sources. However, these methods face challenges in resolving conflicts between model knowledge and retrieved information, as highlighted by Shi et al. (2023). Additional work in this area includes efforts by Dziri et al. (2021), Peng et al. (2023), and Li et al. (2023c), who explore various techniques for grounding LLM outputs in external knowledge sources.

**Generation and Fine-tuning Strategies:** Researchers have also developed strategies to reduce hallucinations during the generation process or through model fine-tuning. Lee et al. (2022) introduced factual-nucleus sampling to balance output diversity and factual accuracy. Reinforcement learning from human feedback (RLHF) has been employed by Ouyang et al. (2022) and Touvron et al. (2023) to align LLMs with desired criteria, including truthfulness. Other approaches include careful curation of instruction-tuning data (Zhou et al., 2023) and linguistic calibration techniques (Mielke et al., 2022). Recent work by Tian et al. (2024) has further explored fine-tuning strategies specifically targeting factuality improvement.

**Leveraging the Latent Space:** An emerging area of research investigates the internal representations of LLMs to understand and manipulate their behaviour. Studies have suggested the existence

of a "truthfulness" direction in the latent space of these models. For example, Li et al. (2023b) proposed Inference-Time Intervention to identify and modify factuality-related directions in model activations. Azaria & Mitchell (2023) introduced SAPLMA, suggesting that LLMs may have an internal awareness of their own inaccuracies. This line of inquiry has been further developed by Burns et al. (2023), who explored methods for discovering latent knowledge, and Marks & Tegmark (2023), who examined the geometry of truth representations in LLMs. Additional insights have been provided by Subramani et al. (2022) and Zou et al. (2023), who have explored techniques for understanding and manipulating the internal representations of these models. Kossen et al. (2024) further propose to directly predict the semantc entropy using probes acting on different hidden layers of the LLM.

## 8 CONCLUSION

This work introduces two novel approaches for uncertainty quantification in large language models: Semantic Embedding Uncertainty (SEU) and its amortized version (ASEU). Our methods leverage semantic embeddings to achieve more robust and nuanced estimations of semantic uncertainty compared to existing techniques. While SEU provides improved accuracy over traditional approaches, ASEU offers a significant computational advantage by enabling uncertainty estimation in a single forward pass. This efficiency is particularly crucial for real-time applications and when dealing with larger language models where multiple forward passes can be prohibitively expensive. Empirical evaluations across multiple datasets and frontier LLMs demonstrate that our embedding-based methods provide more accurate uncertainty quantification than traditional approaches, particularly in scenarios where minor linguistic variations or additional correct information might lead to overestimation of uncertainty. Furthermore, ASEU's ability to maintain comparable performance to SEU while drastically reducing computational overhead represents a substantial step towards making uncertainty quantification more practical and accessible in production environments.

While our results are promising, several limitations of this work should be acknowledged. Our experimental setup primarily focused on short-answer questions and responses, which may not fully capture the complexity and diversity of real-world LLM applications that often involve longer, more nuanced responses. The use of Rouge-L score as an automatic evaluation metric, while suitable for short answers, may not be appropriate for assessing longer or more complex responses. This limitation restricts the generalisability of our findings to broader LLM use cases. Additionally, while we used multiple datasets, they were all in the domain of question-answering. The effectiveness of our methods on other types of language tasks, such as code generation, remains to be explored. Our study also focused on a specific set of commonly used open source LLMs, and the performance and behaviour of our methods on larger models were not investigated.

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

## A    APPENDIX

### A.1    ROC CURVES

We provide the full ROC curves of the methods considered in the main text, across various models and evaluation datasets.

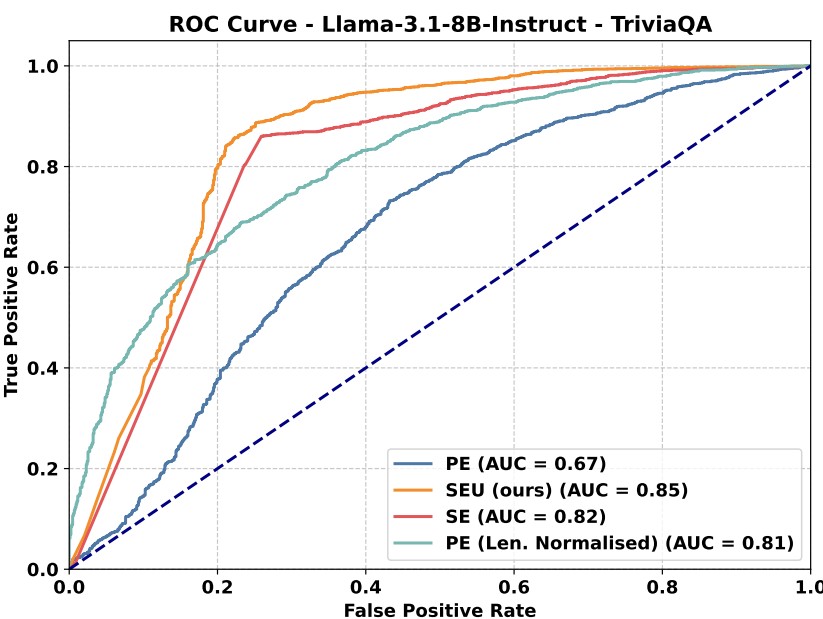

Figure 4: ROC Curve for Llama-3.1-8B-Instruct model on TriviaQA dataset

### A.2    MODEL PROMPTS AND EXAMPLE RESPONSES

This section presents the prompts used for each model and provides examples of their responses to demonstrate the brevity of the generated answers.

#### A.2.1    PROMPTS

For the initial experiments, we used the following prompts:

- Llama and Phi models: "Answer the following question as briefly as possible"
- Mistral model: "Answer the following question briefly using a few words"

After fine-tuning, we used the same prompts except for the Llama model, where we changed the prompt to: "Give a short reply to the following question".

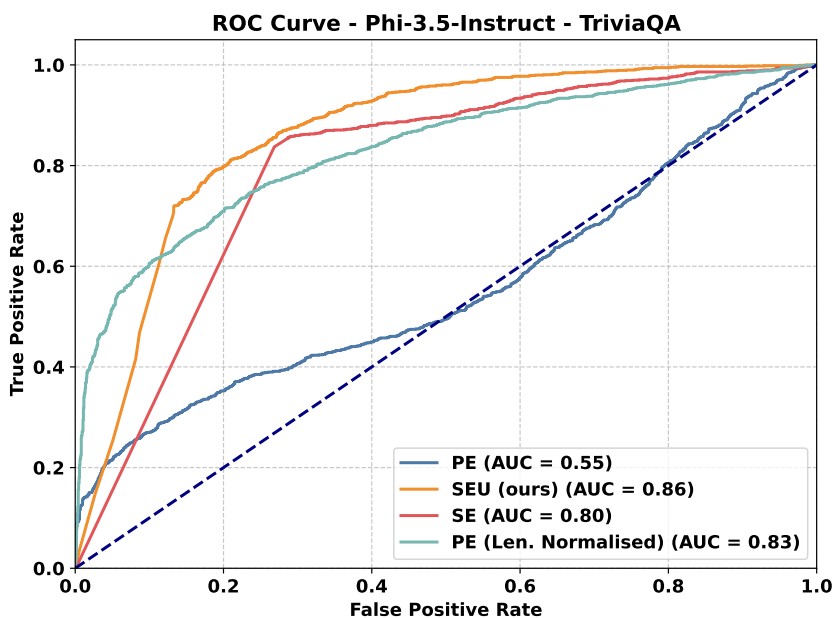

Figure 5: ROC Curve for Phi-3.5-Instruct model on TriviaQA dataset

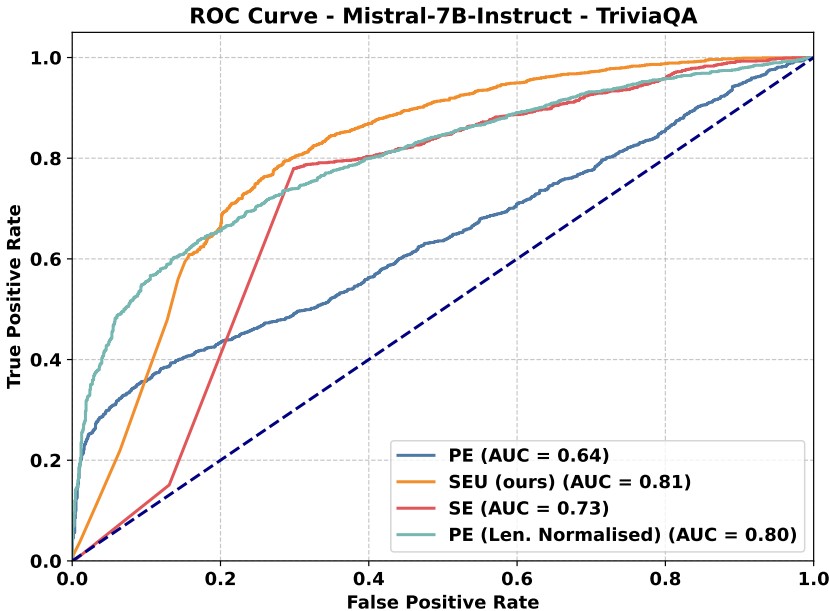

Figure 6: ROC Curve for Mistral-7B-Instruct model on TriviaQA dataset

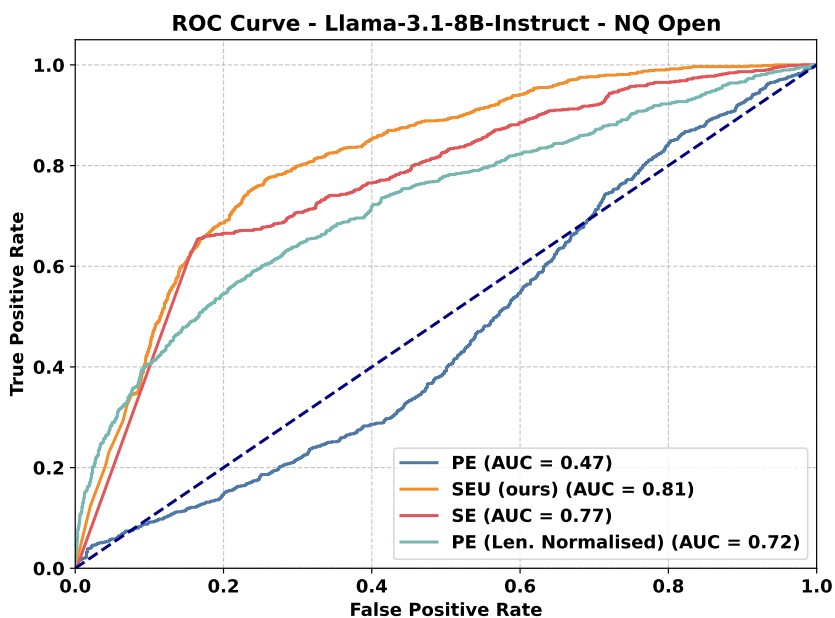

Figure 7: ROC Curve for Llama-3.1-8B-Instruct model on NQ Open dataset

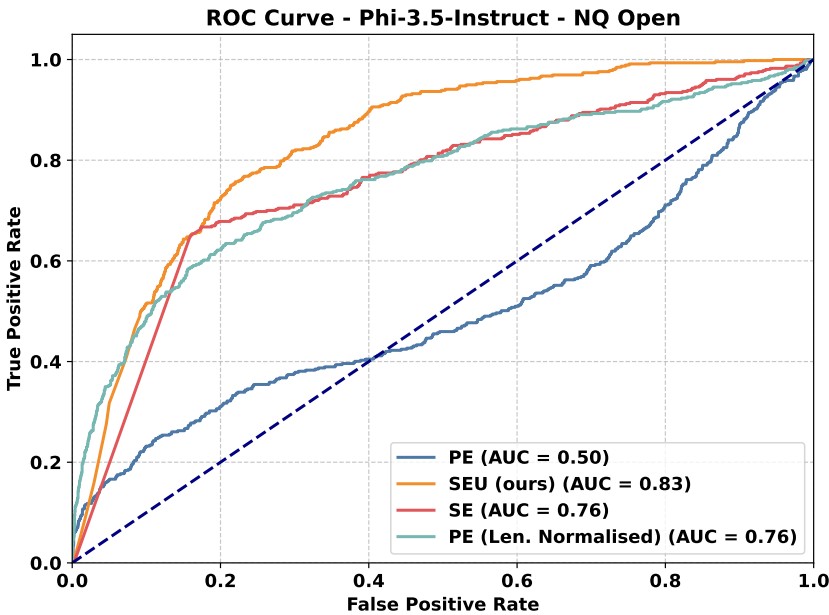

Figure 8: ROC Curve for Phi-3.5-Instruct model on NQ Open dataset

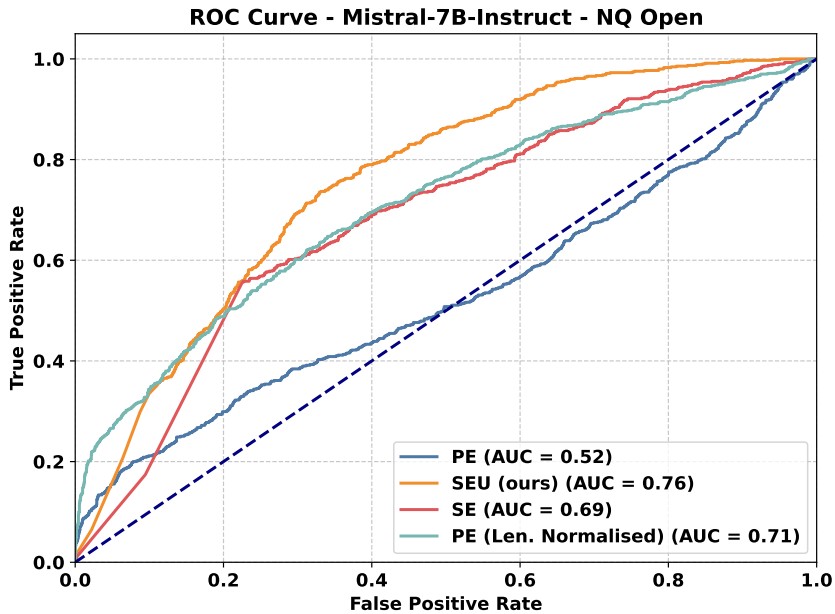

Figure 9: ROC Curve for Mistral-7B-Instruct model on NQ Open dataset

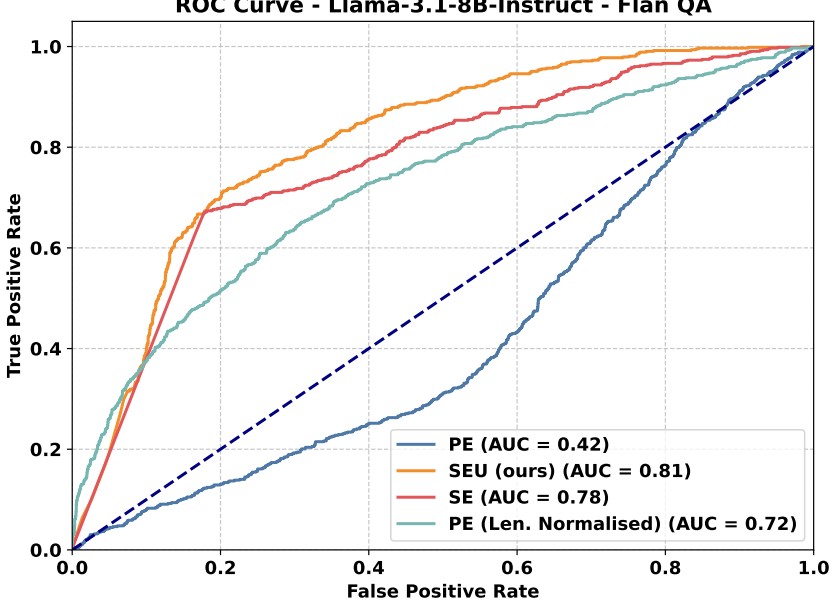

Figure 10: ROC Curve for Llama-3.1-8B-Instruct model on Flan QA dataset

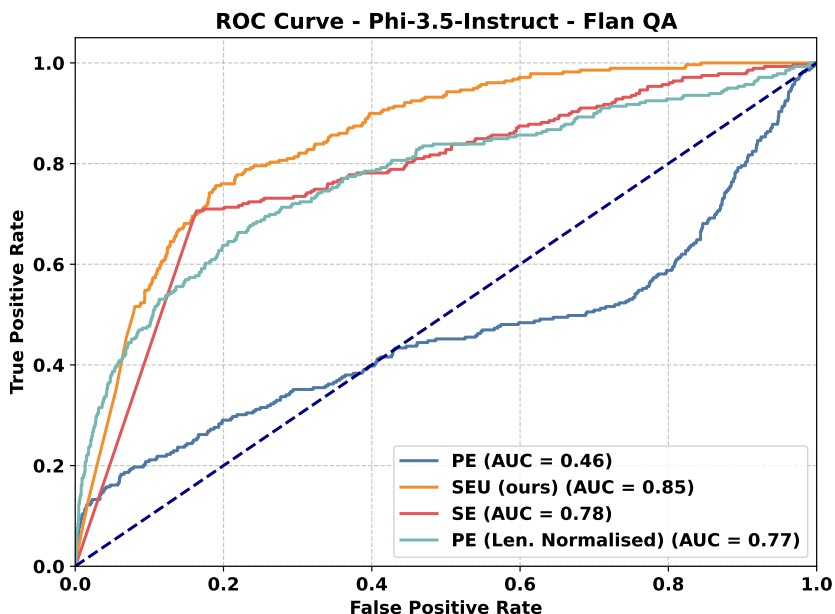

Figure 11: ROC Curve for Phi-3.5-Instruct model on Flan QA dataset

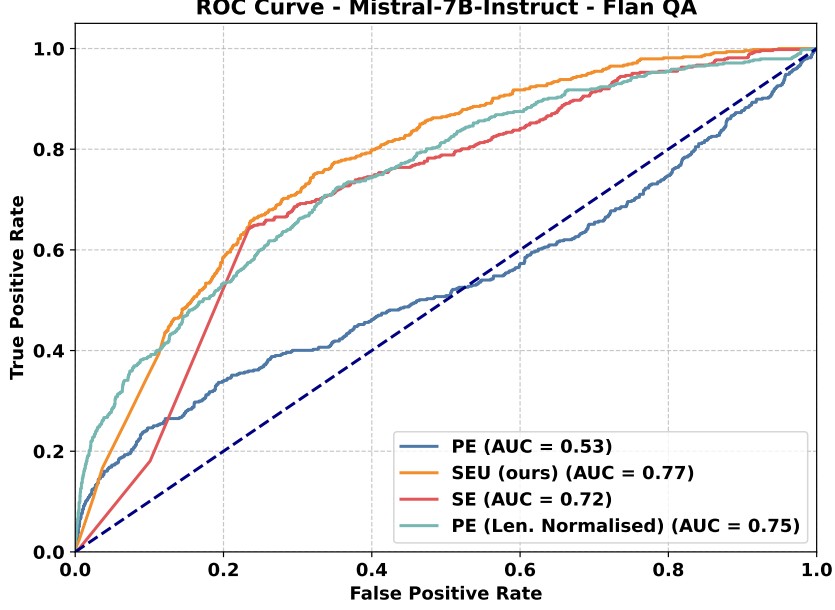

Figure 12: ROC Curve for Mistral-7B-Instruct model on Flan QA dataset

A.2.2 EXAMPLE RESPONSES

Tables 4 and 5 include two examples from the NQ Open dataset to demonstrate that the models indeed generate short answers, adhering to the instruction for brevity.

Table 4: Example responses from different models to the question "When was the first Hunger Games book published?"

| Model | Responses |
|-------|-----------|
| Llama | <ul><li>September 14, 2008</li><li>September 14, 2008</li><li>September 14, 2008.</li><li>September 14, 2008.</li><li>September 14, 2008</li></ul> |
| Mistral | <ul><li>2008 (The Hunger Games)</li><li>2008 (The Hunger Games)</li><li>2008 (The Hunger Games)</li><li>2008</li><li>2008 (The Hunger Games)</li></ul> |
| Phi | <ul><li>2009</li><li>2009</li><li>2009</li><li>2009</li><li>2009</li></ul> |

In the first example, while Llama and Mistral provide the correct publication year (2008) for "The Hunger Games", Phi consistently gives an incorrect year (2009). This illustrates both the models' ability to generate brief responses and the potential for factual inaccuracies in their outputs.

In the second example, all models correctly identify Exodus 20 as a location for the Ten Commandments. Llama consistently provides both locations (Exodus and Deuteronomy), while Mistral and Phi primarily focus on Exodus. This demonstrates the models' capacity to provide accurate, concise information, with some variation in the level of detail provided.

These examples highlight the instruction tuned models adherence to the brevity instruction while showcasing differences in their knowledge and response patterns.

Table 5: Example responses to the question "What is the location of the Ten Commandments in the Bible?"

| Model | Responses |
|---|---|
| Llama | • Exodus 20:1-17 and Deuteronomy 5:6-21.

• Exodus 20:1-17 and Deuteronomy 5:6-21.

• Exodus 20:1-17 and Deuteronomy 5:6-21.

• Exodus 20:1-17 and Deuteronomy 5:6-21.

• Exodus 20:1-17 and Deuteronomy 5:6-21. |
| Mistral | • Exodus 20 (King James Version)

• Exodus 20 (King James Version)

• Exodus 20:1-17

• Exodus 20 (King James Version)

• Exodus 20 (Old Testament)

• Exodus 20 (King James Version) |
| Phi | • Exodus 20:1-17

• Exodus 20:1-17 in the Old Testament

• Exodus 20:1-17

• Exodus (Exodus 20:1-17) and Deuteronomy (Deuteronomy 5:6-2)"

• Exodus 20:1-17 |

