# OpenReview forum: "Improving Uncertainty Quantification in Large Language Models via Semantic Embeddings"
_ICLR.cc/2025/Conference — Submitted to ICLR 2025_

### Official Review · Reviewer_TjGS · 2024-11-02

**Soundness:** 2
**Presentation:** 3
**Contribution:** 3
**Rating:** 6
**Confidence:** 4

**Summary:**

This paper identifies that bidirectional entailment used for semantic uncertainty is sensitive to minor variations in responses.
To address this issue, the authors propose a refined method, semantic embedding uncertainty (SEU), which is based on the average pairwise cosine similarity of the generated responses’ embeddings. The SEU consistently outperforms other baselines on three short-answer datasets. However, SEU still requires multiple runs of LLMs to collect a set of responses. To estimate uncertainty within a single forward pass, Amortised SEU is presented, which introduces a latent variable model to impute latent semantic embeddings with regard to the embeddings of an input.  Experimental results show ASEU achieves better performance than another single forward-pass baseline, length normalized predictive entropy.

**Strengths:**

1. the paper is well-written with a clear storyline
2. the Amortised SEU is able to estimate uncertainty within a single forward pass,  which is efficient and interesting

**Weaknesses:**

1. Apart from the Amortised SEU, I find that the scientific contribution of SEU (average pairwise cosine similarity) is not sufficient.
First, the proposed SEU is a simple adapted version of the original semantic uncertainty. Second,   The authors claim that bidirectional entailment cannot provide continuous scores, but it is doable to use the probability of the NLI model for this purpose. I would suggest adding another baseline that replaces cosine similarity in SEU with the NLI scores.
2. The experimental setting should be improved. First, it is necessary to compare SEU to other embedding-based methods, e.g., INSIDE: LLMs' Internal States Retain the Power of Hallucination Detection (ICLR 2024). Second, it is useful to report multiple metrics for evaluating uncertainty methods, e.g., Brier and ECE. Last, even though the authors mention the problem of the ROUGE-L metric and only short-answer datasets, I would believe that it is easy to mitigate these issues, e.g., Bert score and evaluation of long-form questions like TruthfulQA
3. Amortised SEU required further fine-tuning while other baselines are unsupervised. The generality of ASEU is not clear

**Questions:**

1. in line 185, "cosine similarity provides a continuous metric". In fact, the bidirectional entailment can also offer continuous scores by using the output probability (or logit).
2. ROUGE-L cannot capture the semantic equivalence well and is insensitive to word orders. The evaluation step needs further improvement by using better metrics such as BERTScore and LLM-as-Judge (though stated in the conclusion)
3. It is important to report multiple metrics for uncertainty estimation, e.g., Brier and ECE.
4. a key baseline is missing: INSIDE: LLMs' Internal States Retain the Power of Hallucination Detection (ICLR 2024), which develops an EigenScore metric to measure the semantic consistency in the embedding space.
5. In Table 2, it is hard to see the benefit of SEU since it leads to a worse FPR

---

> ### Author Response · Authors · 2024-11-21
> **Response to Reviewer TjGS (Part 1)**
>
> We sincerely thank the reviewer for their thoughtful and detailed feedback. We appreciate the opportunity to clarify the concerns.
>
> Q1: In line 185, "cosine similarity provides a continuous metric". In fact, the bidirectional entailment can also offer continuous scores by using the output probability (or logit).
>
> We appreciate the reviewer's insightful comment regarding the potential of using the output probabilities from the entailment classification to express the degree of entailment between different responses. Based on this suggestion, we have included the entailment probability as an additional baseline in our experiments. Furthermore, to enhance the robustness of our evaluation, we use a LLM as a judge (GPT-4o) to assess the correctness of the generated responses.We find that SEU outperforms the average pairwise entailment probability baseline in 8 out of 9 cases across different models and datasets. The following table lists the AUROC values for different uncertainty measurement methods across models and datasets.
>
> | Model | Dataset | Semantic Embedding Uncertainty (SEU) | Semantic Entropy (SE) | NLI Entailment Uncertainty |
> |-------|----------|-------------------------------------|----------------------|---------------------------|
> | Llama | NQ Open | **0.715** | 0.687 | 0.701 |
> |       | Natural Questions | 0.707 | 0.669 | **0.712** |
> |       | TriviaQA | **0.803** | 0.764 | 0.787 |
> | Phi | NQ Open | **0.755** | 0.726 | 0.735 |
> |      | Natural Questions | **0.747** | 0.699 | 0.706 |
> |      | TriviaQA | **0.801** | 0.745 | 0.763 |
> | Mistral | NQ Open | **0.719** | 0.668 | 0.691 |
> |         | Natural Questions | **0.699** | 0.654 | 0.692 |
> |         | TriviaQA | **0.732** | 0.679 | 0.700 |
>
> Q2: ROUGE-L cannot capture the semantic equivalence well and is insensitive to word orders. The evaluation step needs further improvement by using better metrics such as BERTScore and LLM-as-Judge (though stated in the conclusion)
>
> Based on this suggestion, we provide results using a LLM as a judge (GPT-4o) to assess the correctness of the generated responses (please see the above response)
>
> Q3: It is important to report multiple metrics for uncertainty estimation, e.g., Brier and ECE.
>
> We appreciate your suggestion to report multiple metrics for uncertainty estimation, such as the Brier score and ECE. However, in the context of free-form question answering (QA) and natural language generation (NLG), these metrics pose significant challenges due to the open-ended nature of the responses. More specifically, calculating the Brier score and ECE requires estimating the total probability mass assigned to all possible correct answers. In free-form text generation, there are nearly infinitely many ways to express a correct response, making this estimation infeasible. Language models assign probabilities to specific token sequences, not to the underlying semantic meanings. Therefore, these metrics cannot accurately capture uncertainty over the correctness of responses. As noted by Kuhn et al. (2023),
>
> “The AUROC is a better measure of uncertainty for free-form question answering and NLG than calibration measures like the Brier score, which are often used in classification or for multiple choice QA. In order to estimate the Brier score, we would need to estimate the entire probability mass assigned to any possible way of saying the correct answer. This is intractable for free form text where we do not have access to probabilities about meanings.”
>
> Q4: a key baseline is missing: INSIDE: LLMs' Internal States Retain the Power of Hallucination Detection (ICLR 2024), which develops an EigenScore metric to measure the semantic consistency in the embedding space.
>
> We thank the reviewer for the pointer to INSIDE. We will include this baseline in the next revision.
>
> Q5: In Table 2, it is hard to see the benefit of SEU since it leads to a worse FPR
>
> The intent of presenting Table 2 was to dissect the contributions to the AUROC from both TPR and FPR, thereby illustrating where SEU provides benefits over SE. Specifically, we aimed to show that the SEU’s higher AUROC stems from a higher TPR which demonstrates SEU's superior ability to detect when the language model is certain and likely to produce correct answers. In contrast, SE by using bidirectional entailment inflates uncertainty and is susceptible to flagging correct answers as having high uncertainty.
>
> Reference:
>
> Lorenz Kuhn, Yarin Gal, and Sebastian Farquhar. Semantic uncertainty: Linguistic invariances for uncertainty estimation in natural language generation. In International Conference on Learning Representations, 2023

---

> ### Comment · Reviewer_TjGS · 2024-11-24
> **a question about Brier and ECE**
>
> > “The AUROC is a better measure of uncertainty for free-form question answering and NLG than calibration measures like the Brier score, which are often used in classification or for multiple choice QA. In order to estimate the Brier score, we would need to estimate the entire probability mass assigned to any possible way of saying the correct answer. This is intractable for free form text where we do not have access to probabilities about meanings.”
>
> Thanks for your responses. I have a question about the metrics of Brier and ECE. If you can obtain labels for the generated answers, e.g., true or false, it is still possible to compute Brier and ECE. See one reference below:
>
> Just Ask for Calibration: Strategies for Eliciting Calibrated Confidence Scores from Language Models Fine-Tuned with Human Feedback

---

> ### Author Response · Authors · 2024-11-27
> **Response to Reviewer TjGS (Part 2)**
>
> Thank you for your follow-up and for pointing us to the paper "Just Ask for Calibration: Strategies for Eliciting Calibrated Confidence Scores from Language Models Fine-Tuned with Human Feedback." We appreciate your insightful suggestion.
>
> You are correct that if we obtain binary labels (true or false) for the generated answers and elicit confidence scores from the language model, it is possible to compute calibration metrics like the Brier score and Expected Calibration Error (ECE). However, our study focuses on uncertainty estimation methods that do not rely on prompting the model to provide explicit confidence scores.
>
> Moreover, recent research [1] (figure 2) indicates that language models' self-reported confidence scores may not be well-calibrated, especially when limited to a small number of samples per query. This challenges the reliability of calibration metrics derived from such confidence scores.
>
> We acknowledge the value of calibration metrics and agree that incorporating them could provide additional insights. However, integrating these metrics would necessitate a different experimental setup that includes confidence elicitation, which is beyond the scope of our current work. Our aim was to compare uncertainty estimation techniques that operate without additional prompts.
>
> We appreciate your suggestion, as it highlights a valuable avenue for future research.
>
> [1] Measuring short-form factuality in large language models https://arxiv.org/abs/2411.04368.

---

> > ### Comment · Reviewer_TjGS · 2024-11-27
> >
> > > You are correct that if we obtain binary labels (true or false) for the generated answers and elicit confidence scores from the language model, it is possible to compute calibration metrics like the Brier score and Expected Calibration Error (ECE). However, our study focuses on uncertainty estimation methods that do not rely on prompting the model to provide explicit confidence scores.
> >
> > My point here is that it can make claims more solid by reporting multiple metrics, which is a common way to assess uncertainty estimation. I am not asking to compare to prompt-based methods.

---

> ### Author Response · Authors · 2024-12-02
> **Response to reviewer TjGS**
>
> Thank you for your feedback once again. We agree that reporting multiple metrics can strengthen our claims, and we appreciate the suggestion to consider the Brier score and ECE. However, as you've noted, calculating these metrics requires access to the model's probability estimates of correctness for each answer.
>
> In the free-form setting of our paper, the model generates open-ended responses without accompanying confidence scores. The probabilities assigned by the model are over sequences of tokens rather than over the semantic correctness of the answers. To compute the Brier score or ECE, we would need to either:
>
> 1.) Elicit Confidence Scores via Prompting: This would involve modifying our methodology to include prompts that ask the model for its confidence in each answer i.e prompt based methods.
>
> or
>
> 2.) Restructure the Task Format: By converting our free-form questions into multiple-choice or true/false formats, we could use the model's probabilities over these discrete options. However, this would change the nature of our task from free-form generation to a classification problem, hence not requiring an estimate of semantic uncertainty.
>
> Given these constraints, we believe that including Brier score or ECE in our current work isn't feasible without altering our experimental setup. Could you please help us understand if there would be another way of calculating ECE/Brier? We do sincerely appreciate your comments and we believe a bit more clarification would help us improve.

---

### Official Review · Reviewer_8m62 · 2024-11-04

**Soundness:** 2
**Presentation:** 2
**Contribution:** 1
**Rating:** 5
**Confidence:** 4

**Summary:**

This paper introduce, the technical challenges include addressing "hallucinations" in LLMs, where models generate coherent but incorrect responses, and the difficulty of applying traditional uncertainty quantification methods to the open-ended nature of natural language generation. Our innovations introduce Semantic Embedding Uncertainty (SEU) to estimate semantic uncertainty more accurately by leveraging semantic embeddings, and an amortized version (ASEU) that reduces computational costs by estimating uncertainty in a single forward pass.

**Strengths:**

1 The paper presents a novel approach to uncertainty quantification in large language models (LLMs) by introducing Semantic Embedding Uncertainty (SEU) and its amortized version (ASEU). These methods innovatively leverage semantic embeddings to address the limitations of traditional uncertainty estimation techniques, offering a more nuanced understanding of semantic uncertainty without relying on strict bidirectional entailment criteria.
2 The research demonstrates high quality through  empirical evaluation across multiple question-answering datasets and state-of-the-art LLMs. The paper provides a good comparison against existing methods, showcasing SEU's superior performance in accurately quantifying uncertainty.
3 The paper is well-structured and clearly articulates the problem, the proposed solutions, and their significance.

**Weaknesses:**

1. Generalization Limitations: The SEU method proposed in this paper has mainly been experimentally validated for question-answering tasks, and its effectiveness and generalization ability in other types of natural language processing tasks, such as text summarization or machine translation, have not been fully tested.
2. Computational Resource Consumption: Although the amortized SEU (ASEU) reduces computational overhead, during the model training phase, especially when it involves inference and optimization of latent variables, it may still require relatively high computational resources.
3. Dependence on Pre-trained Models: The SEU method relies on high-quality pre-trained language models to generate semantic embeddings. If the pre-trained models themselves have biases or inaccuracies, it could affect the accuracy and reliability of the SEU method.
4. Figure 1,2,3,  model size diversity analyses needed.  While the paper provides a comparison of uncertainty estimation methods across different models, it would be beneficial to include models of varying sizes to get a more comprehensive understanding of how the proposed methods scale. The current selection of models does not fully represent the spectrum of LLMs, particularly the very large models (e.g., 70 billion parameters) or smaller models (e.g., 3 billion parameters). Including a broader range of model sizes could provide insights into the scalability and generalizability of the SEU and ASEU methods.
5 . Close Model Variations analysing needed.  It would be insightful to see comparisons that include close variations of the same model architecture but with different sizes. This could help elucidate whether the performance of the uncertainty estimation methods is more dependent on the model architecture or its size. Such an analysis could reveal trends that are crucial for understanding the behavior of SEU and ASEU in different practical scenarios.

**Questions:**

1. Generalization to Other NLP Tasks?  The SEU method has been validated primarily for question-answering tasks. Could the authors discuss the potential effectiveness and generalization of the SEU method to other natural language processing tasks such as text summarization or machine translation, where the context and requirements might differ significantly?

2. Computational Resources in Model Training? While ASEU is praised for reducing computational overhead during inference, what are the computational resource requirements during the model training phase, particularly when dealing with the inference and optimization of latent variables? How does this compare to traditional methods in terms of efficiency?

3. Reliability on Pre-trained Models? The SEU method's performance seems to heavily rely on the quality of the pre-trained language models for generating semantic embeddings. How sensitive is the SEU method to potential biases or inaccuracies in these pre-trained models, and what measures can be taken to mitigate such issues?

4. Diversity in Model Size Analysis? Figure 1, along with other figures, compares uncertainty estimation methods across a limited range of model sizes. Could the authors expand on the analysis to include a more diverse set of model sizes, especially very large models (e.g., 70 billion parameters) and smaller models (e.g., 3 billion parameters), to provide a comprehensive understanding of how the proposed methods scale and perform across different model sizes?

5. Analysis of Close Model Variations? It would be insightful to see an analysis that includes close variations of the same model architecture but with different sizes. How does the performance of the uncertainty estimation methods vary with model size while keeping the architecture constant? What trends can be observed that would help us understand the behavior of SEU and ASEU in practical applications with different model sizes?

---

> ### Author Response · Authors · 2024-11-21
> **Response to Reviewer 8m62 (Part 1)**
>
> We sincerely thank the reviewer for their feedback. We appreciate the opportunity to clarify the concerns.
>
> Q1: Generalization to Other NLP Tasks?
>
> In our work, we specifically focus on free-form question-answering (QA) tasks for hallucination detection, following the precedent set by prior research in this area (e.g., [1,2,3]). Free-form QA is particularly susceptible to hallucinations due to the open-ended nature of the task and the lack of constraints compared to some other tasks. Our goal was to develop and validate the SEU method within this context, where accurately estimating semantic uncertainty can have a significant impact on the reliability of language model outputs. While our experiments and validations are centred on QA tasks, we believe that the underlying principles of SEU and ASEU —quantifying semantic uncertainty through comparing response embeddings — are applicable to other NLP tasks.
>
> Q2 (a): Computational Resources in Model Training?
>
> While ASEU introduces a fine-tuning phase to learn the latent semantic representations, we designed the process to be efficient and practical for real-world applications. Specifically: we fine-tuned the ASEU model using 20,000 question-answer pairs from the TriviaQA dataset. The finetuning process only requires learning the additional adapter weights, while the LLM weights are held fixed. For this fine-tuning step, we used a single NVIDIA A100 GPU. The fine-tuning step can be done in less than 4 hours.
>
> Q2 (b) On your question about How does this compare to traditional methods in terms of efficiency?
>
> Could you please elaborate which traditional methods you are referring to in this question?
>
> Q3 Reliability on Pre-trained Models?
>
> While we acknowledge that the quality of semantic embeddings could impact SEU's performance. As we show in our experiments, the off-the-shelf sentence-BERT model delivers strong performance across a variety of datasets. The modular nature of our approach allows for easy substitution of the embedding model component. As better sentence embedding models become available, they can be readily incorporated to further improve performance.
>
> To support this point, we provide additional experiments where we use NV-embed which is a new SOTA sentence embedding model [5].
>
> | Model | Dataset | Semantic Embedding Uncertainty (SEU) [NV-EMBED] | Semantic Entropy (SE) |
> |--------|----------|----------------------------------|-------------------|
> | Llama-8B | NQ-open | **0.835** | 0.772 |
> |        | TriviaQA | **0.843** | 0.820 |
> | Phi 3.5B| NQ-open | **0.851** | 0.756 |
> |       | TriviaQA | **0.848** | 0.808 |
>
> The consistent performance we observe across different datasets and LLMs and 2 sentence embedding models, provides strong evidence for the robustness of our method.
>
> Q4 Diversity in Model Size Analysis?
>
> We appreciate the reviewer's suggestion about including a broader range of model sizes. However, we believe our current analysis already provides strong evidence for the effectiveness of our method across a meaningful spectrum of model sizes
>
> - Phi-3.5B (~3.5 billion parameters)
> - Mistral-7B (7 billion parameters)
> - Llama-8B (8 billion parameters)
>
> Following the reviewers suggestion we have added new models up to 27B parameters. The new models added are:
>
> - Gemma-2-9B  (9 billion parameters)
> - Gemma-2-27B (27 billion parameters)
>
> | Model | Dataset | Semantic Embedding Uncertainty (SEU) | Semantic Entropy (SE) | NLI Entailment Uncertainty |
> |---------|----------|-----------------------------------|---------------------|--------------------------|
> | Gemma-2-9B | NQ Open | **0.793** | 0.742 | 0.776 |
> | Gemma-2-27B | NQ Open | **0.788** | 0.751 | 0.765 |
>
> The consistent trends we observe across the current range of model sizes (3.5B to 27B) provide evidence that our method's effectiveness is not heavily dependent on model sizes.
> While experimenting with 70B+ models would be interesting, our current analysis focuses on model sizes that are readily accessible to most researchers. Please note that it is generally expensive for many academic researchers to experiment with models around or beyond the 70B parameter range.
>
> Q5: Analysis of Close Model Variations?
>
> We thank the reviewer for the suggestion. We have now added results from the gemma family of models [4], we have focused on 9B and 27B to show the consistency of our approach across size variations of the same model architecture. Please see the results in the response above.

---

> > ### Author Response · Authors · 2024-12-02
> > **Response to Reviewer 8m62 (Part 2)**
> >
> > References:
> >
> > [1] Lorenz Kuhn, Yarin Gal, and Sebastian Farquhar. Semantic uncertainty: Linguistic invariances for uncertainty estimation in natural language generation. In International Conference on Learning Representations, 2023
> >
> > [2] Nikitin, A., Kossen, J., Gal, Y., & Marttinen, P. (2024). Kernel Language Entropy: Fine-grained Uncertainty Quantification for LLMs from Semantic Similarities.
> >
> > [3] Jie Duan, Haotian Cheng, Shihao Wang, Chengwei Wang, Aidana Zavalny, Ronghui Xu, Bhavya
> > Kailkhura, and Kaidi Xu. Shifting attention to relevance: Towards the uncertainty estimation of
> > large language models
> >
> > [4] Gemma: Open Models Based on Gemini Research and Technology. Gemma Team. https://arxiv.org/abs/2403.08295
> >
> > [5] NV-Embed: Improved Techniques for Training LLMs as Generalist Embedding Models https://arxiv.org/abs/2405.17428

---

> ### Author Response · Authors · 2024-11-27
> **Rebuttal Reminder**
>
> Dear Reviewer 8m62,
>
> We appreciate that you are likely to be reviewing multiple other papers; however, as we are approaching the end of the discussion period (less than one week remaining), we would greatly appreciate your feedback on our rebuttals. We appreciate your feedback and we hope that we have adequately addressed the concerns you raised. We have also included additional experiments based on your suggestions including bigger models.
>
> Your additional insights would be valuable in helping us improve the paper further.
>
> Kind regards, The authors of paper 10173

---

### Official Review · Reviewer_vHa8 · 2024-11-06

**Soundness:** 2
**Presentation:** 3
**Contribution:** 2
**Rating:** 6
**Confidence:** 3

**Summary:**

This paper addresses the problem of quantifying uncertainty in large language models. Specifically, previous methods, such as Semantic Entropy (SE), use bidirectional entailment criteria to assess whether two responses share the same meaning. However, the authors argue that bidirectional entailment criteria are sensitive to linguistic variations. To address this, they propose SEU, which quantifies uncertainty using cosine similarities between the semantic embeddings of different responses. Additionally, since both SE and SEU require multiple forward passes, the authors introduce an amortized version of SEU that models the underlying semantics as latent variables, requiring only a single forward pass while still achieving strong results.

**Strengths:**

1. **Important problem**: The paper focuses on an important and valuable problem: quantifying the uncertainty of large language models.

2. **Efficient method**: The authors propose an amortized version of SEU that models the underlying semantics as latent variables. This approach requires only a single forward pass but still demonstrates very strong performance.

3. **Experimental validation**: The authors conduct experiments on three QA datasets across three LLMs, demonstrating the effectiveness of the proposed SEU and ASEU methods.

**Weaknesses:**

1. **Reliability of cosine similarity of embeddings as an absolute measure of semantic relatedness**: I have some concerns about using cosine similarity as an absolute measure to describe the similarity between two responses. As shown in [1], the embedding space is isotropic, meaning that some texts are very close together within a narrow cone, while others are far apart. Thus, cosine similarity may not function well as an absolute measure.

2. **Advantages of SEU compared to SE**: The authors claim that SE models are sensitive to linguistic variability and minor wording differences, whereas SEU models are more robust and focus on the underlying semantic content. However, the SE method uses NLI classifiers that take the combined input of both responses, allowing for more fine-grained interaction between them. In contrast, SEU encodes responses independently, lacking such fine-grained interaction. As such, I find the argument for SEU being superior to SE unconvincing. Additionally, the experiments are conducted on a single model for both SE and SEU, which may not reveal whether the observed advantages are due to model choice or differences in methodology.

3. **Bidirectional entailment criterion can only act as a binary measure**: The authors claim that bidirectional entailment criteria can only yield binary outcomes. However, since the criterion relies on an entailment classification, the output probability of the entailment class has the potential to express the degree of relatedness between different responses.

[1] Jun Gao, Di He, Xu Tan, Tao Qin, Liwei Wang, Tie-Yan Liu. Representation Degeneration Problem in Training Natural Language Generation Models. ICLR 2019.

**Questions:**

Please see my comments on the Weakness section.

---

> ### Author Response · Authors · 2024-11-21
> **Response to Reviewer vHa8 (Part 1)**
>
> We sincerely thank the reviewer for their thoughtful and detailed feedback. We appreciate the opportunity to clarify the concerns about our proposed Semantic Embedding Uncertainty (SEU).
>
> On Weakness #1: **Reliability of cosine similarity of embeddings as an absolute measure of semantic relatedness**
>
> We understand the concern about the isotropy of the embedding space, as discussed in Gao et al. [1], which can affect the reliability of cosine similarity measures. However, we would like to clarify that the representation degeneration problem highlighted in [1] primarily pertains to language generation models trained with likelihood loss, where the goal is to predict the probability of the next word. This can lead to embedding degeneration (of word embeddings), making cosine similarity an unreliable measure of semantic relationships. In contrast, our method employs Sentence-BERT [2], which is specifically designed to produce semantically meaningful sentence embeddings that are suitable for comparison using **cosine similarity**.
> Specifically, Sentence-BERT uses siamese and triplet network structures to fine-tune BERT models, optimising their embeddings for semantic similarity tasks. As stated in [2]: "We use siamese and triplet network structures to derive semantically meaningful sentence embeddings that can be compared using **cosine-similarity**." Therefore we believe that this training approach can alleviate the issue of using cosine similarity which may arise when using embeddings from regular language models.
>
> On Weakness #2: **Advantages of SEU compared to SE**
>
> We appreciate the reviewer's concern regarding the interaction between responses in NLI-based methods versus our SEU approach. While NLI classifiers in SE consider both responses jointly, we argue in this work that the strict entailment criteria they rely on can be overly stringent for measuring semantic uncertainty. More specifically, for sentence A to entail sentence B, all the information in B must be present in A. This is a stringent criterion that does not tolerate additional or missing details, even if they are minor (we provide examples of such failure cases in Table 1.) This strictness means that even minor variations, paraphrasing, or inclusion of extra correct information can cause semantically very similar responses to be classified as non-equivalent, inflating the estimated uncertainty of SE. As we show in Table 2 in the paper: SE overestimates uncertainty as demonstrated by its significantly lower TPR. Lower TPR suggests that SE is classifying more cases as uncertain, even when the model’s response is correct.
>
> On your concern about the experiments: To clarify, our experiments in the paper were conducted using different language models of varying sizes, trained by different research teams (Llama 8B, Mistral 7B, Phi 3.5B, Gemma-2-9B, Gemma-2-27B) to ensure that our findings are robust and not specific to a single model. In addition, for the entailment model used in the Semantic Entropy (SE) baseline, we used the DeBERTa-Large model, consistent with prior work proposing SE Kuhn et al. (2023), Nikitin et a. (2024). We also adopted the same evaluation methodology as these previous papers, using the ROUGE-L score to assess the correctness of generated responses and the Area Under the Receiver Operating Characteristic curve (AUROC) for performance comparisons. Using the same NLI model and the evaluation methodology as in these prior works ensures a fair comparison between SE and our proposed SEU method.
>
> On Weakness #3: **Bidirectional entailment criterion can only act as a binary measure**
>
> We appreciate the reviewer's insightful comment regarding the potential of using the output probabilities from the entailment classification to express the degree of entailment between different responses. Based on this suggestion, we have included the entailment probability as an additional baseline in our experiments. Furthermore, to enhance the robustness of our evaluation (following reviewer TjGS’s suggestion), we use a LLM as a judge (GPT-4o) to assess the correctness of the generated responses. We find that SEU outperforms the average pairwise entailment probability baseline in 8 out of 9 cases across different models and datasets. The following table lists the AUROC values for different uncertainty measurement methods across models and datasets.

---

> ### Author Response · Authors · 2024-11-21
> **Response to Reviewer vHa8 (Part 2)**
>
> | Model | Dataset | Semantic Embedding Uncertainty (SEU) | Semantic Entropy (SE) [DeBERTa Large] | NLI Entailment Uncertainty [DeBERTa Large] |
> |-------|----------|-------------------------------------|----------------------|---------------------------|
> | Llama 8B | NQ Open | **0.715** | 0.687 | 0.701 |
> |       | Natural Questions | 0.707 | 0.669 | **0.712** |
> |       | TriviaQA | **0.803** | 0.764 | 0.787 |
> | Phi 3.5B | NQ Open | **0.755** | 0.726 | 0.735 |
> |      | Natural Questions | **0.747** | 0.699 | 0.706 |
> |      | TriviaQA | **0.801** | 0.745 | 0.763 |
> | Mistral 7B | NQ Open | **0.719** | 0.668 | 0.691 |
> |         | Natural Questions | **0.699** | 0.654 | 0.692 |
> |         | TriviaQA | **0.732** | 0.679 | 0.700 |
>
>
> | Model | Dataset | Semantic Embedding Uncertainty (SEU) | Semantic Entropy (SE) | NLI Entailment Uncertainty |
> |---------|----------|-----------------------------------|---------------------|--------------------------|
> | Gemma-2-9B | NQ Open | **0.793** | 0.742 | 0.776 |
> | Gemma-2-27B | NQ Open | **0.788** | 0.751 | 0.765 |
>
>
> [1] Jun Gao, Di He, Xu Tan, Tao Qin, Liwei Wang, Tie-Yan Liu. Representation Degeneration Problem in Training Natural Language Generation Models. ICLR 2019.
>
> [2] Nils Reimers and Iryna Gurevych. Sentence-BERT: Sentence embeddings using Siamese BERTnetworks. In Proceedings of the 2019 Conference on Empirical Methods in Natural Language
>
> Lorenz Kuhn, Yarin Gal, and Sebastian Farquhar. Semantic uncertainty: Linguistic invariances for
> uncertainty estimation in natural language generation. In International Conference on Learning
> Representations, 2023
>
> Nikitin, A., Kossen, J., Gal, Y., & Marttinen, P. (2024). Kernel Language Entropy: Fine-grained Uncertainty Quantification for LLMs from Semantic Similarities.

---

> ### Author Response · Authors · 2024-11-27
> **Rebuttal Reminder**
>
> Dear Reviewer vHa8,
>
> We appreciate that you are likely to be reviewing multiple other papers; however, as we are approaching the end of the discussion period (less than one week remaining), we would greatly appreciate your feedback on our rebuttals. We appreciate your detailed feedback and we hope that we have adequately addressed the concerns you raised. We have also included additional experiments based on your suggestions.
>
> Your additional insights would be valuable in helping us improve the paper further.
>
> Kind regards, The authors of paper 10173

---

> > ### Comment · Reviewer_vHa8 · 2024-11-30
> > **Response to Authors**
> >
> > Thank you for your detailed and elaborate response, as well as the experiments conducted. I sincerely apologize for the late reply.
> >
> > (1) I agree that cosine similarity is a meaningful objective. However, I still have some minor concerns that there is no evidence to suggest cosine similarity is well-calibrated. From what I understand about how Sentence-BERT is trained, there doesn’t appear to be evidence supporting its direct use as an absolute value. For instance, while we can infer that A is more similar to B than C because sim(A, B) > sim(C, B), can we reliably say that A and B are highly similar if their similarity score exceeds 0.9? Is there evidence to support this interpretation? I would greatly appreciate clarification from the authors on this point.
> >
> > (2) I believe this is more of a problem of NLI rather than the architecture. Regarding the experiments, I was referring specifically to different NLI classifiers rather than LLMs. While I agree with the authors' decision to follow the settings of previous works, given that SE and SEU are notably different methods, I think incorporating some additional NLI classifiers could provide valuable insights.
> >
> > (3) I sincerely appreciate the authors’ efforts and would be very interested to see how the methods perform when using different NLI classifiers. This could further validate the superiority of SEU.

---

> ### Author Response · Authors · 2024-12-02
> **Response to reviewer vHa8**
>
> Once again we thank the reviewer for their thoughtful question and recommendations.
>
> Q1.) There doesn’t appear to be evidence supporting cosine similarity's direct use as an absolute value.
>
> While we agree that absolute cosine similarity values require careful interpretation, we would like to draw attention to Sentence-BERTs strong performance on the STS benchmark dataset (Cer et al., 2017), which has human labels of semantic similarity on a 0-5 scale. In this dataset, for a given pair of sentences, scores are defined with the following criteria [Table 1  (Cer et al., 2017)]:
>
> Score 5: Complete semantic equivalence (e.g., 'The bird is bathing in the sink' vs 'Birdie is washing itself in the water basin')
>
> Score 4: Mostly equivalent with minor differences (e.g., 'Two boys on a couch are playing video games' vs 'Two boys are playing a video game')
>
> Score 3: Roughly equivalent with important differences (e.g., 'John said he is considered a witness but not a suspect' vs '"He is not a suspect anymore." John said')
>
> Score 2: Not equivalent but sharing some details
>
> Score 1: Not equivalent but same topic
>
> Score 0: Completely dissimilar
>
> The sentence-bert model achieves a strong Spearman correlation of 0.85 on this benchmark [1], demonstrating that higher absolute values of cosine similarity correspond to higher human-assessed semantic similarity. While this correlation doesn't guarantee that specific cosine similarity thresholds map directly to the 0-5 scale, it provides strong evidence that high absolute values of cosine similarity (for eg: greater than 0.9) will generally correspond to high semantic similarity assigned by humans.
>
> Q2.) I think incorporating some additional NLI classifiers could provide valuable insights.
>
> Once again we thank the reviewer for their valuable suggestion to help improve our work. We have now added Roberta Large as an additional NLI classifier. We are also commencing experiments on XLNet, but please note that these experiments can take a fairly long time (due to the large size of these datasets) and we may not be able to finish it in the remaining time for the discussion period (~1 day left). We will make sure to add these experiments into the main PDF once we are allowed to do so.
>
> | Model | Dataset | Semantic Embedding Uncertainty (SEU) | Semantic Entropy (SE) [Roberta Large] | NLI Entailment Uncertainty [Roberta Large] |
> |--------|----------|----------------------------------|-------------------|------------------------|
> | Llama 8B | NQ-open | **0.715** | 0.698 | 0.711 |
> | | TriviaQA | **0.803** | 0.781 | 0.790 |
> | Phi 3.5B | NQ-open | **0.755** | 0.694 | 0.722 |
> | | TriviaQA | **0.801** | 0.729 | 0.755 |
>
> Q3) would be very interested to see how the methods perform when using different NLI classifiers
>
> Please see the response above to Q2.
>
> References:
>
> Daniel Cer, Mona Diab, Eneko Agirre, Iigo LopezGazpio, and Lucia Specia. 2017. SemEval-2017
> Task 1: Semantic Textual Similarity Multilingual
> and Crosslingual Focused Evaluation. In Proceedings of the 11th International Workshop on Semantic
> Evaluation (SemEval-2017), pages 1–14, Vancouver, Canada.
>
> [1] Nils Reimers and Iryna Gurevych. Sentence-BERT: Sentence embeddings using Siamese BERTnetworks. In Proceedings of the 2019 Conference on Empirical Methods in Natural Language

---

> > ### Comment · Reviewer_vHa8 · 2024-12-02
> > **Response to Authors**
> >
> > The answer to Q1 is not fully convincing to me. Regarding the STS task, we might encounter a scenario where, for three pairs with human-annotated similarity scores of (3, 4, 5), a model predicts cosine similarities of either (0.1, 0.2, 0.3) or (0.91, 0.92, 0.93). Both predictions would yield the same Spearman’s rank correlation with the human-annotated scores. In this context, strong STS performance may not provide sufficient evidence that cosine similarity can be directly interpreted as an absolute value.
> >
> > Thank you for conducting the experiments to demonstrate the superiority of SEU. Using cosine similarity of embeddings to relax the bidirectional entailment criterion provides interesting and valuable insights. As such, while I remain unconvinced by the response to Q1, I have increased my score to reflect these contributions.

---

### Author Response · Authors · 2024-11-25
**General Response**

We sincerely thank all reviewers for their thoughtful feedback and valuable suggestions, which have been instrumental in improving the quality of our work. We are encouraged that the reviewers appreciated the quality of writing and the efficiency benefits of our proposed ASEU method.

However, we acknowledge that some reviewers may have had questions regarding the proposed SEU method and its advantages. We have addressed these points comprehensively in our individual responses, along with the additional concerns and questions raised by each reviewer.

Furthermore, we have significantly enhanced our experimental evaluation following the reviewers' suggestions, including the addition of a new baseline and expanding our tests to include larger models up to 27B parameters. We believe these improvements help clarify the paper's main contributions while strengthening the overall quality of our experimental results.

---

### Meta-Review · Area_Chair_9juS · 2024-12-23

**Metareview:**

This paper is interested in improving uncertainty quantification for large language models. The idea is to obtain better estimates by integrating an improved version of semantic entropy. This permits taking into account a more accurate notion of uncertainty compared to either (i) just trivially examining how different responses are or (ii) using bidirectional entailment. The authors introduce some improved mechanisms and then study the performance of these approaches, including an efficient version that is less computationally expensive. The main idea is to use cosine similarity between emebddings.

As strong points, the paper provides a very simple solution that directly offers improvements. This is always nice to see.

As weaknesses, my thoughts are the following. First, this paper studies a relatively narrow area; effectively, it is searching for mechanisms to improve a component of semantic entropy. To do so, the authors propose the most straightforward approach. This is not by itself bad, but it does make me think there is much more to do and learn in this direction.

As also noted by the reviewers, cosine similarity can be tricky to reliably use. This is not a problem because there are a ton of alternatives (or combinations) that could be used here, and I would like to see more and understand the pros and cons. In fact, I would love to see a combination of SE with the authors’ idea, since they should be complementary, each handling cases the other does poorly on. In addition, evaluation was a good start but could be far more extensive (and this should be relatively easy).

Ultimately I felt that the paper was on the right track, but there’s much more to add to make it a strong paper that passes the bar.

**Additional Comments On Reviewer Discussion:**

I mostly agreed with the reviewers that had questions on the use of cosine similarity; while the authors did write a strong rebuttal, ultimately it would be good to study all of this more comprehensively for the next iteration.

---

### Decision · Program_Chairs · 2025-01-22

Reject